# The Effect of Situational Experiment Conditions on Hasty Decision Making in the ‘Beads Task’

**DOI:** 10.3390/brainsci13020359

**Published:** 2023-02-19

**Authors:** Kristoffer Klevjer, Gerit Pfuhl

**Affiliations:** 1Department of Psychology, Faculty of Health Sciences, UiT–The Arctic University of Norway, N-9037 Tromsø, Norway; 2Department of Psychology, Faculty of Social and Educational Sciences, Norwegian University of Science and Technology, N-7491 Trondheim, Norway

**Keywords:** demand characteristics, experimenter bias, information sampling, jumping to conclusions, delusion

## Abstract

‘Jumping to Conclusions’, or hasty decision making, is widely studied within clinical and computational psychology. It is typically investigated using the ‘beads task’, a sequential information sampling paradigm, and defining one or two draws as jumping to conclusion. Situational experimental conditions, e.g., group vs. individual testing, abstract vs. cover story, show-up fee or course credit, frequently vary between studies. Little effort has been dedicated to investigating the potential effects of demand characteristics on hasty decision making. We explored this in four samples of participants (*n* = 336), in different situational experiment conditions, with two distinct variations of the beads task. An abstract ‘Draws to Decision’ (DtD) variant, and a cover story combined DtD and probabilistic inferences variant. Situational conditions did not have a significant effect on overall DtD for either variant. However, when using ‘extreme scores’ (DtD of 1 or 1 to 2) as a measure of hasty decision making, situational conditions had an effect for the abstract variant, with individual testing having the fewest hasty decision makers (DtD1: Mann–Whitney *U* = 2137.5, *p* = 0.02; DtD1-2: Mann–Whitney *U* = 2017.5, *p* < 0.01), but not for the cover story variant. Our results suggest that the abstract variant is more susceptible to test conditions, especially if a categorisation is used to classify hasty decisions. This does not imply that the cover story variant is better suited to capturing jumping to conclusions behaviour, but highlights the importance of mirroring the situational conditions between different samples. We recommend that testing conditions should be fully disclosed.

## 1. Introduction

“Jumping to Conclusions” or hasty decision making, is widely researched within both clinical psychology and computational psychiatry [1,2,3,4]. The most common paradigm for investigating Jumping to Conclusions behaviour is the “beads task”. The task was originally used to study probabilistic (Bayesian) inference, and was later modified into a “Draws to Decision” task [5,6]. Of clinical relevance, persons with delusions or who are delusion prone are often observed to make ‘hasty’ decisions, i.e., sample less evidence before making a decision, compared to controls [7,8,9,10,11]. Despite Draws to Decision being a widely used task for assessing group differences, little effort has been dedicated to investigating the role of demand characteristics on this task. For example, test conditions for patients and healthy control groups do not always match or are frequently not reported [12]. In this meta-analysis of 35 studies, only three studies clearly reported the test conditions and 13 studies provided no information at all, and only two studies reported the complete method of reimbursment. We here set out to explore the role of a set of contextual factors on sampling behaviour.

In a typical version of this task, participants are presented with two different urns filled with coloured beads. Each bead is in one of two possible colours (e.g., either red or blue), and the two urns differ in terms of the ratio between these bead colours (e.g., 0.85/0.15 in one urn, 0.15/0.85 in the other). One of the urns is then chosen at random, and participants gather information by drawing beads, one at a time, replacing the observed evidence. In a probabilistic inference version, participants are asked to provide their probability estimates as they observe the evidence [6]. These estimates can then be compared in relation to other concepts (such as across mental disorders) and/or compared to an ideal Bayesian observer.

The beads task was later modified into the ‘Draws to Decision’ (DtD) variant [5]. In this variant, participants are free to sample information until they feel confident enough to decide from which urn they believe to be drawing from (quantifying their ‘Draws to Decision’). Highly delusion-prone participants have been found to make hastier decisions in this task variant (as displayed by either an overall lower Draws to Decision score, or a higher group proportion of ‘extreme’ Draws to Decision scores of 1–2 or 1 [7,8,9,10,11,13,14]).

The mechanism(s) behind this (somewhat inconsistent) display of hasty decision making in the delusion prone remains elusive. To mitigate misunderstanding as a cause for Jumping to Conclusions behaviour [15], different variations of the beads task have been developed. Some studies include multiple trials, vary the difficulty level (e.g., 0.60/0.40-ratio [2]), combine a measurement of Draws to Decision with probabilistic inferences, and/or provide ‘cover stories’. For example, to decrease the level of abstraction and potentially increase task comprehension, urns have been replaced with ponds and beads with fish [16,17]. Furthermore, to mitigate working memory demands often found in clinical groups [1], the ratio of the two urns or ponds remain on screen.

The task repeatedly yields group-level differences between the delusion prone (e.g., persons with a schizophrenia diagnosis) and controls. However, hasty decision making and the individuals’ severity of delusion proneness within the groups are not related [7,8,9,10,11]. More problematically, test conditions between the patient and control group may vary and can potentially contribute to group differences. Across studies, the execution of the task varies too. This might contribute to the reported inconsistency of the Jumping to Conclusions behaviour, although differences in the characteristics of the patient group may also play a non-negligible role.

It is ethically, timewise and economically too costly to assess demand characteristics in large patient groups. However, the role of demand characteristics such as using an abstract versus a cover story variant, or testing individually or in groups, and with or without the option to ask the experimenter, can be assessed in non-clinical samples. Another advantage of using a non-clinical sample is that compared to a clinical sample we expected participants to have sufficient working memory and verbal intelligence. This in turn is required to allow a fair comparison between the abstract and cover story version [1]. The cover story may add comprehension, but the surplus information may burden a person with low working memory capacity. Indeed, the abstract version is rarely presented solely textually. Already, the first studies used images to illustrate the ratio of the two urns [5,6], which should help in comprehension.

Situational factors could affect participants’ behaviour, e.g., by allowing implicit learning and observation for group tested participants, or increased motivation or effort in individually tested participants. Various factors may also affect motivation, e.g., the rationale given for playing the task, including it being for course credit or for helping patients. Motivation, in turn, may affect the effort spent in comprehending the task instructions. We think it plausible that the abstract version of the beads task is more susceptible to these situational factors even if experimenter bias is kept low in abstract and cover story variants [12]. It is also not known whether testing in a group setting will be beneficial or detrimental to task performance. This is hard to predict, as the beads task on the one hand requires understanding the task instructions, but on the other hand has no single correct answer [18]. We here set out to explore the role of these situational factors on sampling behaviour. 

### Aims

Sampling behaviour might depend on demand characteristics and or motivation, such that an abstract Draws to Decision variant is more susceptible to situational conditions (e.g., testing individually vs. in groups) than a cover story ‘fish task’ variant in which both probabilistic inferences and Draws to Decision were assessed. We report data from four individual samples of a homogeneous non-clinical population. As will be described further below (see Section 2.2), the cover story version took approximately the same amount of time for all participants, irrespective of their Draws to Decision. This is not the case in the abstract version, where “hasty decision makers” would more quickly proceed to the next task. This may matter in group testing, but not for individual testing. Hence, we expected more heterogeneity or a larger influence of situational context in the abstract version. This susceptibility should be reflected in larger variation of the sampling behaviour in group vs. individual testing in the abstract version.

## 2. Materials and Methods

### 2.1. Participants and Ethics

Four independent samples of participants (*n* = 336) were recruited, all students at UiT—The Arctic University of Norway.

For sample 1, 71 participants (34 female, 34 male, 3 non-binary/undisclosed, 18 to 36 years old; Mean: 24.08, Median: 23, SD: 3.795, 75% IQR: 22–26) were recruited via e-mail, posters, and convenience sampling. They varied in area of study, with 14 (19.7%) being psychology students. Data collection was carried out from June to December 2018. These participants were individually tested in a testing room equipped with a computer at the university. The experimenter left the room after instructions were given. Task instruction was provided both on screen and on a printed hand-out.

For sample 2, 77 participants (54 female, 22 male, 1 non-binary/undisclosed, 19 to 34 years old, Mean: 22.24, Median: 21, SD: 2.917, 75% IQR: 20.5–23) were recruited via an undergraduate course in cognitive psychology. They were mainly psychology students. Data collection was carried out in February 2020. These participants were tested in smaller groups (6–10 per group; seated with a minimum of 2 m distance between participants) in a computer pool at the university. In each session, a brief instruction was given to all via a beamer, and more detailed instructions were provided through the computer program/on screen, as for sample 1. The experimenter would answer questions and help with technical issues if needed.

For sample 3, 136 participants (89 female, 43 male, 4 non-binary/undisclosed, 19 to 44 years old; Mean: 23.24, Median: 22, SD: 4.189, 75% IQR: 21–24) were recruited from two undergraduate courses: introduction to psychology and cognitive psychology. They were mainly psychology students. Data collection was carried out from February to April 2021. These participants were tested in smaller groups (6–10 per group; seated with a minimum of 2 m distance between participants) in a computer pool at the university. Detailed instructions were provided on screen prior to the start of the task (see Figure 1B), and a summarized version was provided on screen at each piece of evidence/step of the task (see Figure 1C).

For sample 4, 79 participants (62 female, 14 male, 3 non-binary/undisclosed, 20 to 37 years old, Mean: 22.72, Median: 22, SD: 2.767, 75% IQR: 21–24) were recruited via an undergraduate course in cognitive psychology. They were mainly psychology students. Data collection was carried out in February 2022. These participants were tested remotely (online) in digital testing sessions. The instructions stressed that they have to be seated alone and undisturbed. They received a random show-up time (six different times in total) at which they were provided with the link to the experiment, and told to complete the experiment at that time. Instructions were provided on screen as for sample 3 (see Figure 1B,C).

All participants were blinded in that they only knew the experiment was set in cognitive psychology. To the best of our knowledge, no prior course(s) had included any information relating to neither Jumping to Conclusions behaviour nor the beads task or any other aspects relevant for information sampling tasks. All core task instructions were provided on screen, and experimenters were instructed to stick to these instructions and not provide alternative explanations, thus minimizing experimenter bias [12].

The four experiments were parts of a broader project investigating information sampling, and in all four studies other tasks were administered too. Thus, the overall experiment durations were between 0.75 and 1.2 h. Participants in sample 1 received a show-up fee of 150 NOK (~15 USD). Participants in samples 2, 3 and 4 received fulfilment of a course requirement. Note, participation in the study was one of several potential ways to obtain this, and participants were in addition given the opportunity to opt-out of having their data being used for research (one in study 3 and one in study 4). All participation was voluntary.

Inclusion criteria were: 18 to 50 years of age, normal or corrected to normal eyesight, no neurological disorder, no drug use within three months prior to the testing session (excluding caffeine, nicotine, and alcohol), and no current intake of central nervous system medications (e.g., antidepressants, antiepileptics, or ADHD medications). The participants read and signed an informed consent form prior to the experiment, and were provided with the opportunity to contact an experimenter should they have any questions, or wish to withdraw from the experiment. This also included the online study, where students took the test during specified hours and could call or email the experimenter.

All studies obtained ethical approval from the local institutional ethics board at the department of psychology, UiT—The Arctic University of Norway.

### 2.2. Materials: The Beads Task(s)

Figure 1 presents the two task variants. Samples 1 (individually tested) and 2 (group tested) received an abstract ‘Draws to Decision’ variant of the beads task including multiple trials, while samples 3 (group tested) and 4 (individually tested remotely) received a single trial of a cover story variant of the beads task (see Figure 1).

#### 2.2.1. Abstract ‘Draws to Decision’ Variant

The abstract beads task is a computerized task that measures participants’ amount of sequential information sampling prior to a binary decision (‘Draws to Decisions’ variant) [5].

Participants were presented with two different urns filled with beads, each bead in one of two possible colours, and with the two urns differing in the ratio between these colours. One of the urns was chosen at random, and participants were asked to sequentially draw beads (with replacement of the observed evidence), until they felt confident to decide from which of the two urns they believed they were drawing from. The participants received the first piece of evidence and could then request additional information by pressing ‘space’, with the newest piece of evidence displayed on screen along with the previously obtained evidence (see Figure 1A). Once sufficiently confident, they indicated their belief by pressing ‘1’ for ‘Urn with X% majority colour A’ or ‘9’ for ‘Urn with X% majority colour B’. Upon choosing, the trial was terminated, without feedback, and the next trial was presented.

The participants completed four trials. The first two with opposite 0.85/0.15-ratio urns, and the latter two with opposite 0.60/0.40-ratios. For sample 1, an asymmetrical fifth trial was added, with one 0.85/0.15 urn and one 0.50/0.50 urn. Unbeknownst to the participants, the observed sequences of evidence were predetermined, and identical across participants (per trial). For trial 1, the exact sequence was ‘YYYYYYYYYBYYYYYYYYYY’ with ‘Y’ standing for yellow beads and ‘B’ standing for blue beads (see Figure 1A). A visual representation of the ratio of the two urns was displayed in the upper left corner (Figure 1A), reducing working memory (assuming the participant looks at the information). 

The task was implemented and administered using PsychoPy 3 [16].

#### 2.2.2. Cover-Story Combined Variant

The cover story variant is mathematically equal to that of the abstract Draws to Decision variant, but with three notable differences.

First, rather than being presented with urns and beads, participants were observing a sports fisherman catching fish (each fish in one of two potential colours), and were asked to make a decision as to which of two ponds (A and B; with opposite distributions of coloured fish, 80:20 or 20:80, Figure 1B) he was catching them from [17,19]. Second, participants were asked after each caught fish (piece of evidence) to report the current probability that the fisherman was fishing from pond A or B. Third, unbeknownst to the participants, even after having made a decision, they had to go through all 10 pieces of evidence (maximum amount) and keep making probability estimates and decisions. The latter should reduce differences in how long it took to play the task, i.e., irrespective of the Draws to Decision the task would last as similar amount of time and provide similar information on the screen. 

Participants received the first piece of evidence, were asked to report their probability estimates, and then chose if they felt confident enough to make a decision (either ‘Yes, I chose pond A’, or ‘Yes, I chose pond B’) or needed more evidence (‘No, I need to see more fish’). This process was repeated through all 10 pieces of evidence, with the newest piece of evidence displayed on screen along with the previously obtained evidence (see Figure 1C). There was one trial of this task, and unbeknownst to the participants the observed sequence of evidence was predetermined, and identical across participants. The exact sequence was ‘XXXOXXXXOX’ with ‘X’ being grey fish and ‘O’ being green fish. For each trial the ratio of the two ponds was provided in brackets (see Figure 1C), reducing working memory demands (assuming the participant reads the information).

The task was administered through Qualtrics (Provo, UT, USA).

### 2.3. Procedure

Participants received the invitation and were provided with a test time. They read and signed an informed consent form and were then provided with written instructions on screen at the start of the experiment.

For samples 1 and 2, half of the participants received the abstract Draws to Decision beads task first, while the other half received another information sampling task first (‘the box task’ [20]), and then the beads task. For sample 3, half of the participants received the cover story variant of the beads task first, while the other half received another information sampling task first (a dice throwing task), and then the beads task. For sample 4, half of the participants received the ‘box task’ first, and then the cover story variant of the beads task, while the other half received the dice throwing task first, and then the beads task.

### 2.4. Analysis

In samples 1 and 2, the number of beads sampled prior to making a decision was taken as their ‘Draws to Decision’. In samples 3 and 4, the number of fish sampled prior to making a decision was taken as their ‘Draws to Decision’. The probability estimates at the time of decision are not analysed here. To reduce potential learning effects and/or decrease in miscomprehension over multiple trials, we focus on the first trial in each variant and for each sample. Clinical studies also commonly use only the first trial [12]. Variation in sampling behaviour is assessed by calculating the Coefficient of Variation and using the asymptotic test for the equality of the coefficient of variation [21].

## 3. Results

Descriptive statistics can be seen in Table 1.

The effect of the situational conditions (individual or group) on overall DtD was not significant for either the abstract DtD variant (Mann–Whitney *U* = 2874.5, *p* = 0.25) or the cover-story with both probabilistic inferences and DtD (Mann–Whitney *U* = 4340.5, *p* = 0.41). When using extreme scores (DtD of 1 or 1 to 2) as a measure for hasty decision making, situational conditions had an effect for the abstract DtD variant, with individual testing having the fewest hasty decision makers (DtD1: Mann–Whitney *U* = 2137.5, *p* = 0.02; DtD1-2: Mann–Whitney *U* = 2017.5, *p* < 0.01), but not for the cover-story with combined DtD and probabilistic inferences variant (DtD1: Mann–Whitney *U* = 3681, *p* = 0.16; DtD1-2: Mann–Whitney *U* = 3846.5, *p* = 0.50). Note, sample 1, abstract and tested individually, was also the only sample receiving reimbursement for playing a variety of tasks instead of course credit. A chi-square test comparing hasty (fewer than three beads or fish sampled) and non-hasty participants yielded a significant effect, Χ^2^ = 7.707, *p* = 0.006, Cramer’s V = 0.1524.

The cumulative proportion of participants who had decided as the task progressed, along with the ideal Bayesian observer evidence can be seen in Figure 2. The figure also illustrates some invariance in the testing condition (individual vs. group) to removing hasty participants in the cover story variant but not in the abstract condition. 

Regarding variation in the sampling behaviour, the asymptotic test yielded a significant difference between the four studies, test statistic = 12.91, *p* = 0.0048, and sample 2 had the largest coefficient of variation (see Table 1). However, when limiting sampling to 10 (samples 1 and 2, setting more than 10 beads to 11), the asymptotic test was not significant, *p* = 0.168.

## 4. Discussion

Our results indicate that the abstract Draws to Decision variant is more susceptible to demand characteristics, particularly situational condition. The effect of the situational conditions is especially evident if one uses ‘extreme scores’ (DtD of 1 or 1 to 2) as classification for hasty decision-making behaviour, as is often done within clinical research using this task paradigm [13,14]. As shown in Figure 2A, participants tested in groups were more likely to decide on the very first piece of evidence compared to the participants tested individually in the abstract variant. However, if they continued to sample, the individually tested made their decisions earlier in the trial compared to the group tested. No such pattern was visible for the participants in the cover story variant.

The effect of this response pattern (or lack thereof) was also visible in the variability of sampling between participants within the different conditions. Overall, the largest coefficient of variation could be observed for the group-tested participants using the abstract variant, and this was significantly higher than the other conditions. This is an interesting observation, as low between-participant variation is important for tasks designed to test group differences (as is often the goal in much of the mentioned clinical research), whereas the opposite, namely high between-participant variation is important for tasks designed to test individual differences (as is often the goal in research into cognitive aspects of information sampling and decision making) [22]. An increase in between-participant variation due to situational conditions could therefor make the task less suitable to capture group effects, while simultaneously not making it more suited to capture individual differences, as the increase in variation is not due to meaningful differences, but rather because of an extraneous variable.

One plausible explanation for this situational conditions effect in the abstract Draws to Decision variant is social factors such as assuming good performance is achieved by making quick decisions and or using visual cues whether peers have completed the trial. Despite not being able to directly see the screen of another participant, other cues might be available to infer whether others were still sampling (e.g., keyboard tapping), and this might influence one’s own sampling behaviour. The combined cover story variant is naturally slower-paced (due to the probabilistic inference reporting), and there were far fewer cues (visual or audible) for the group-tested participants indicating how much information others gathered before making a decision.

Importantly, this does not necessarily imply that the abstract Draws to Decision variant is less suited to study Jumping to Conclusions behaviour compared to the cover story variant, only that it is more vulnerable to the experimental condition. Individually tested participants in the abstract variant had a pattern of responding that is most in line with what one would expect regarding the objective degree of evidence (i.e., most sensitive to the rapid decline of additional information per new piece of evidence [23], see Figure 2). This is notable, because collecting evidence in the abstract variant is faster and easier than in the cover story variant, and with a decreased cost of sampling, one would expect these participants to sample more, not less than in the cover story variant to reach an equilibrium between their subjective cost of sampling and their desired level of confidence [24,25].

Overall, in both task variants, and in all experimental situation conditions, including in the sample with monetary show-up compensation, some participants were ‘hasty decision makers’ (as classified by a threshold of 1 or 1–2 pieces of information sampled prior to deciding, as is common in clinical research using this task paradigm [13,14]). Thus, all combinations could be able to capture group differences in “Jumping to Conclusions” when comparing clinical populations to neurotypical controls. However, our results indicate that the impact of situational conditions and extraneous variables differs for the different variants, with the abstract variant being especially prone, and potentially provide a noisier measurement. Highlighting the importance of mirroring not only task parameters between different groups of comparison, but also extraneous variables such as situational conditions and method of reimbursement.

### Limitations

The present research aimed to investigate the potential effects of situational conditions on two distinct variants of the beads task. Importantly, this does not allow for inferences on the two variants’ ability to reliably quantify individual differences. As noted by Hedge et al. [22], many of the upsides of established behavioural tasks (such as the beads task) in use on group-level differences, such as low variability between participants within-group, are what makes them poorly suited for research on individual differences. Despite the task variants yielding variation in sampling behaviour this is not sufficient to conclude on their suitability for individual differences research. Using only the first trial might be more noisy [9]. However, noise might be what differs between patients and healthy controls at the group level [26]. It is possible that the impact of situational factors becomes lower when calculating Draws to Decision scores based on multiple trials. However, this can be confounded by learning, and the appeal of this task is in measuring a decision given some prior information and new evidence. 

In the current paper, we used rather homogeneous and healthy participants in all samples, to isolate the effect of the situational experiment condition [12]. We note that we did not verify that with having measured the participants working memory capacity, verbal intelligence. However, even the presence of extraneous variable effects does not exclude the possibility of a true between-group effect when comparing the delusional-prone and controls. It only highlights the importance of evaluating this aspect when designing future studies.

## 5. Conclusions

Our results highlight the importance of mirroring not only task parameters, but also extraneous variables, such as situational testing condition and method of reimbursement between different samples for comparison. This is especially important when studying potential group-level effects between a clinical and non-clinical group, and when these effects can be expected to be rather modest. Lastly, the potential impact of these extraneous variables also highlights the importance of transparently and completely reporting these in future research and publications. Together, these two recommendations should allow for better insight into the somewhat elusive relationship between hasty decision making and delusional proneness, or at a minimum control for or exclude the potential effect of these extraneous artifacts when researching this delicate relationship.

## Figures and Tables

**Figure 1 brainsci-13-00359-f001:**
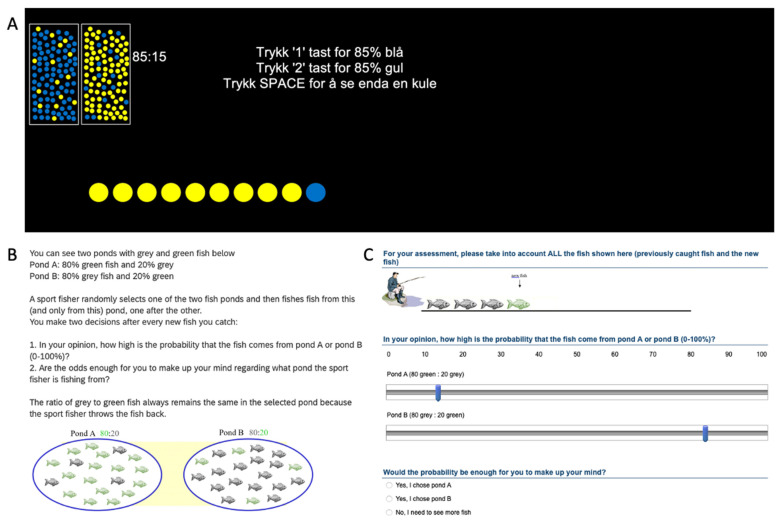
Overview of the two ‘beads task’ variants used. (**A**) Mid-trial for the abstract variant (administered in both Norwegian and English depending on the participant’s native language). Trykk 1/2 task for 85% blå/gul = press 1/2 for 85% blue/yellow, Trykk space for å se enda en kule = Press space to see another bead. Working memory demand was eliminated by providing the ratio of the two urns in the upper left corner. (**B**) Instruction page and pond presentation for the cover story variant. (**C**) Fourth trial in the cover story variant showing first the observation, then the probabilistic judgement, separate for both ponds and finally asking for a decision. Working memory demand was eliminated by providing the ratio of the pond in brackets. Note that the ratio information during a trial is visual in the abstract version but textual in the cover story version.

**Figure 2 brainsci-13-00359-f002:**
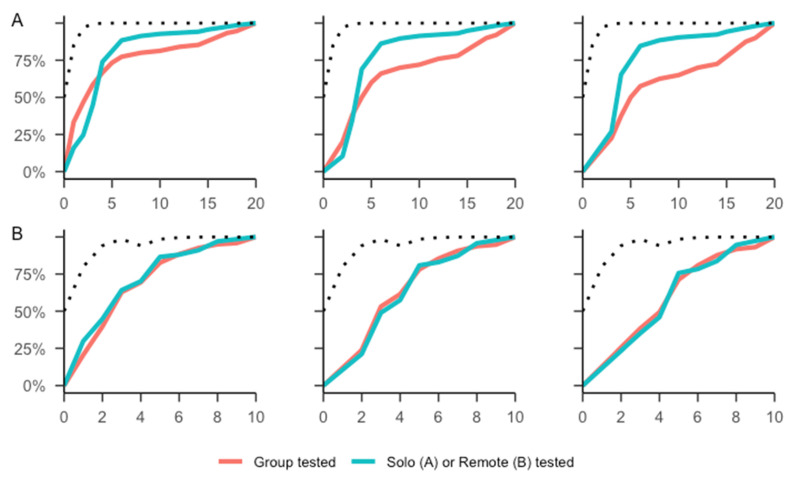
Cumulative proportion of participants having made a decision across the sequence of evidence (in red: group testing (samples 2 and 3); in blue solo/remote testing (samples 1 and 4)). (**A**) Abstract Draws to Decision variant. (**B**) Cover story variant. Left: all participants. Middle: with DtD = 1 participants excluded. Right: with DtD = 1 or 2 participants excluded. Ideal Bayesian Observer thresholds as black dotted lines.

**Table 1 brainsci-13-00359-t001:** Descriptive statistics per condition/sample.

Sample	Participants (Valid)	Mean (Median)	SD (CoV)	Min/Max	DtD1	DtD1-2
Abstract–Individual Abstract–Group	70 (69)	4.49 (4)	3.98 (0.89)	1/20	16%	25%
	75 (75)	5.31 (3)	5.90 (1.11)	1/20	33%	47%
Cover–Group Cover–Remote	124 (121)	3.53 (3)	2.35 (0.66)	1/10	21%	40%
	67 (67)	3.31 (3)	2.37 (0.71)	1/10	30%	45%

Note. Mean (Median), Standard Deviation (SD), Coefficient of Variation (CoV), minimum (Min) and maximum (Max) ‘Draws to Decision’ per sample, for participants (with valid data points). Proportion of the sample with ‘extreme responses’/JtC, DtD of 1 (DtD1) or DtD of 1 or 2 (DtD1-2).

## Data Availability

Data, analysis script (presented analyses, presented table and figure, and ideal Bayesian observer calculation), software for the abstract beads task, and materials for the cover-story beads task are openly available at OSF (https://osf.io/dz7yj/, accessed on 21 December 2022).

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
