# Peer review of "The Effect of Situational Experiment Conditions on Hasty Decision Making in the ‘Beads Task’"

_brainsci, 2023, doi:10.3390/brainsci13020359_

Round 1

Reviewer 1 Report

This paper does a great job at highlighting the significant external noise that can impact proper measurement. The JTC task has many flaws and the authors do an excellent job at detailing these issues while still acknowledging the benefit in attempting to measure these constructs. 

There are some concerns regarding this manuscript that can be addressed in future iterations:

1. the use of the phrase "costly" is used several times to indicate that certain samples may be difficult to recruit and facilitate through research trials. There are minor notes that help clarify the use of this phrasing, but a sentence early on in the introduction to clarify this phrase would be beneficial for readability. 

2. The authors examine several potential circumstances that can impact the results of any clinical trial, but it would be beneficial to acknowledge the following in the limitations/discussion: The abstract and the cover story versions of the JTC task have different denominators in potential evidence shown. This should not have an impact on the DTD variable as participants are generally not instructed on how many pieces of evidence they will see - however this can impact the standard deviations of the samples, that the authors later note to be informative. I think the observation is interesting and still worth noting, but this caveat has to be noted since at least a few individuals in the abstract group declined to make a decision till the very end (Max = 20) which will skew these data more than the cover story task. 

Author Response

This paper does a great job at highlighting the significant external noise that can impact proper measurement. The JTC task has many flaws and the authors do an excellent job at detailing these issues while still acknowledging the benefit in attempting to measure these constructs. 

Thank you

There are some concerns regarding this manuscript that can be addressed in future iterations:

  1. the use of the phrase "costly" is used several times to indicate that certain samples may be difficult to recruit and facilitate through research trials. There are minor notes that help clarify the use of this phrasing, but a sentence early on in the introduction to clarify this phrase would be beneficial for readability. 

Thank you very much for requesting this clarification. We have specified more clearly what is meant by costly, e.g. line 70ff

  1. The authors examine several potential circumstances that can impact the results of any clinical trial, but it would be beneficial to acknowledge the following in the limitations/discussion: The abstract and the cover story versions of the JTC task have different denominators in potential evidence shown. This should not have an impact on the DTD variable as participants are generally not instructed on how many pieces of evidence they will see - however this can impact the standard deviations of the samples, that the authors later note to be informative. I think the observation is interesting and still worth noting, but this caveat has to be noted since at least a few individuals in the abstract group declined to make a decision till the very end (Max = 20) which will skew these data more than the cover story task. 

Thank you very much for raising this issue. Indeed, the asymptotic test became non-significant if sampling was limited to 10 beads (coding more sampling as 11 beads requested). We report this now in the result section. We have rewritten the discussion and note that even for role of the Coefficient of Variation sampling should be similar, i.e. same amount of beads.

Reviewer 2 Report

1.     The authors suggested, “…But with no clear, or only a weak relationship, between this hasty decision-making and the individuals’ severity of delusion-proneness within the groups…”, yet the research didn’t aim to solve this issue?  

2.     It’s unclear why the authors recruited four independent samples of participants? How the authors control the demographics of different group of participants? There is unequal group size, and what’s the rational in assigning the participants/ group of participants into different situations. 

3.     How to ensure the psychology students didn’t know about JTC biases before the experiment? 

4.     It’s unfair comparison for group 1 to receive compensation while group 2-4 are getting credits for the course

5.     Why sample 1 and 2 were assigned to do the abstract DTD variant of the beads task while sample 3 and 4 received a cover-story version? Why not all of them going through all the conditions? 

6.     What’s the differences in the counterbalance between sample 1,2,4 with sample 3? Please elaborate. (Page 5 line 198 – Page 6 line 203) 

7.     The results can be further elaborated. The authors can do further analyses to support the stated hypotheses, how the situation or the experimental condition affects the participants’ performance in different version of the tasks. 

8.     The discussion is not well supported with the references. Please further elaborate how the results of the current study contribute to the topic of JTC. 

Author Response

Reviewer 2

  1. The authors suggested, “…But with no clear, or only a weak relationship, between this hasty decision-making and the individuals’ severity of delusion-proneness within the groups…”, yet the research didn’t aim to solve this issue?  

Thank you, we have rewritten the section to make more clear that we are not studying individual differences.

  1. It’s unclear why the authors recruited four independent samples of participants? How the authors control the demographics of different group of participants? There is unequal group size, and what’s the rational in assigning the participants/ group of participants into different situations. 

Thank you for asking this. We have rewritten the introduction, highlighting that many studies do not report test conditions which might though affect hasty decision making. We now cite Klein et al. 2012 on demand characteristics.

  1. How to ensure the psychology students didn’t know about JTC biases before the experiment? 

 This is a very good question. We know what the students had in their curriculum but not outside. We still are highly confident that they were not familiar with the JTC. None of the participants has told us after the experiment that they thought we measure the JTC. We are aware that this is not full-proof and wished we had asked them afterwards what they think the experiment was about (as done in Coles et al. 2022 a multi-lab test of the facial feedback hypothesis)

  1. It’s unfair comparison for group 1 to receive compensation while group 2-4 are getting credits for the course

Thank you very much for pointing that out. We specified more clearly that this was a show-up fee, not compensation for the task. Using a Chi-square test we find a significant difference. However, we did not counterbalance this, and this coincides with the group also being individually tested.  

  1. Why sample 1 and 2 were assigned to do the abstract DTD variant of the beads task while sample 3 and 4 received a cover-story version? Why not all of them going through all the conditions? 

Thank you for asking this. We were concerned about carry-over effects if we used a within-participant design. However, future studies should consider within-participant designs.

  1. What’s the differences in the counterbalance between sample 1,2,4 with sample 3? Please elaborate. (Page 5 line 198 – Page 6 line 203) 

We have rewritten this section to make more clear that in all four samples participants performed another information sampling task, i.e. there was no difference in counterbalancing 

  1. The results can be further elaborated. The authors can do further analyses to support the stated hypotheses, how the situation or the experimental condition affects the participants’ performance in different version of the tasks. 

Thank you for asking this. We have added the analysis based on show-up fee / reimbursement vs course credit. However, only one study had a show-up fee, so this coincides with being abstract and individual testing. We added the analysis for the Coefficient of Variation.

Other factors, like experimenter bias were reduced to a similar minimum in all four studies. The instructions for sample 1 and 2 were identical. The instructions for sample 3 and 4 were identical. The room was identical for sample 2 and 3. We have performed a chi-square test (data is not normally distributed) for abstract vs cover and individual vs group testing, and we have (see reviewer 1’s remark) checked the analysis for limiting sampling in study 1 and 2 to maximally 10 (coding sampling more as 11 beads requested). Only the latter affected the Coefficient of Variation result and is reported. The other analyses did not change the overall story and for clarity we did not add them.

  1. The discussion is not well supported with the references. Please further elaborate how the results of the current study contribute to the topic of JTC. 

Thank you. We have rewritten the discussion to address your concern.

Reviewer 3 Report

The authors present data across four distinct studies looking into situational factors that might influence hasty decision making on the ‘beads task’. Specifically, the authors conclude that the (original) abstract variant (administered individually) led to the fewest proportion of participants responding hastily (i.e., less DtD of 1 or 2). Investigating situational factors in a very well researched paradigm is commendable, but overall, I am struggling to determine the unique contribution this paper offers the field more generally. Some of the issues I had are listed below.

The tasks used are not unique and appear to be replications of previously used tasks in this literature. This means the current paper is little more than a replication of previous papers – the only (potential) novel aspect was group-based administration, but even this was not purely unique (others have used group-administered beads tasks – see Ryan McKay’s work on the beads task).

Importantly, there was no underlying theory for why these situational factors (i.e., abstract/cover-story; group/solo/remote) could influence DtD, just the notion that they might be important. No hypotheses for the different situational factors were offered.

The authors did not summarise previous studies that have already used the four conditions implemented in this study (i.e., what have other studies found when using solo vs. remote test conditions).

There was no measure of the key individualistic factor discussed at length in the Introduction (i.e., delusion-proneness). This seemed like a missed opportunity to look at the potential interaction between situational and individual-difference factors.

Taking the first trial of any beads task is highly problematic (see McLean et al., 2018), as it so unreliable. This is an important limitation to address.

The Abstract read more like a summary of the Introduction than of the entire study, with key aspects missing (e.g., sample size, tasks used, listing the situational factors).

Author Response

The authors present data across four distinct studies looking into situational factors that might influence hasty decision making on the ‘beads task’. Specifically, the authors conclude that the (original) abstract variant (administered individually) led to the fewest proportion of participants responding hastily (i.e., less DtD of 1 or 2). Investigating situational factors in a very well researched paradigm is commendable, but overall, I am struggling to determine the unique contribution this paper offers the field more generally. Some of the issues I had are listed below.

The tasks used are not unique and appear to be replications of previously used tasks in this literature. This means the current paper is little more than a replication of previous papers – the only (potential) novel aspect was group-based administration, but even this was not purely unique (others have used group-administered beads tasks – see Ryan McKay’s work on the beads task).

Thank you for requesting a clarification. Our contribution is analysing whether demand characteristics and experiment situation influences sampling behaviour. We have more explicitly stated this and cite Klein et al. The articles included in the meta-analyses by Ross et al. rarely report whether testing was individually or in group. For patients we assume testing was individual as this is very common in clinical psychology, but we do not know whether the instructions were given orally, or written. We even know less how participants in the control group were tested. In very few (we only found 1 in the papers cited in the meta-analysis) information whether participants (any group) got reimbursed is clearly stated. In one study, patients but not controls got reimbursed (Lim et al.).

As you correctly remarked, both abstract vs cover, and individual vs group testing has been used. But we are not aware of whether researchers have looked at whether this affects sampling behaviour. We here find that for the cover story testing condition is not affecting sampling behaviour. For the abstract version we find that testing condition is affecting sampling behaviour, particularly if a categorization into hasty vs non-hasty is made (extreme scores).

Importantly, there was no underlying theory for why these situational factors (i.e., abstract/cover-story; group/solo/remote) could influence DtD, just the notion that they might be important. No hypotheses for the different situational factors were offered.

Thank you for pointing that out, and we apologize to have not made it explicit enough. We have rewritten the introduction and cite Klein et al. 2012. Please see also our reply to your previous comment.

The authors did not summarise previous studies that have already used the four conditions implemented in this study (i.e., what have other studies found when using solo vs. remote test conditions).

Thank you for this comment. Unfortunately, we are not aware of studies that directly compared sampling behaviour in the beads task between abstract and cover story. From word of mouth lab vs remote testing is similar, but we would love to know about articles directly testing it and are happy to cite them and compare it to our results.

There was no measure of the key individualistic factor discussed at length in the Introduction (i.e., delusion-proneness). This seemed like a missed opportunity to look at the potential interaction between situational and individual-difference factors.

Thank you for pointing that out. We have rewritten the introduction. Delusional proneness was not measured in all samples. Given meta-analytical studies showing no consistent relation between JtC and delusion-proneness, but more reliable to find group differences, we clarified now why it is important to ensure groups are tested in as similar as possible conditions.

Taking the first trial of any beads task is highly problematic (see McLean et al., 2018), as it so unreliable. This is an important limitation to address.

Thank you. We do address this limitation in the discussion section.

The Abstract read more like a summary of the Introduction than of the entire study, with key aspects missing (e.g., sample size, tasks used, listing the situational factors).

Thank you, we have rewritten the abstract.

Round 2

Reviewer 2 Report

Thanks for inviting me to review the article, titled "The effect of situational experiment condition on hasty decision-making in the ‘Beads task’", here are some of the major comments, 

-       For the last sentence of the abstract, “We recommend that testing conditions and incentives are reported.”, is it “should be reported” instead of “are reported” ?

-       The authors should state the hypotheses based on the evidence from the existing research in the introduction section

-       The authors should report the mean age of the participants instead of the range. I just wonder how many of the participants were mature students. It may affect the results found as maturity it’s a potential factor affecting “jumping to conclusion” as well. 

-       Will it be valid to compare the results from the four-study given study 1 only has around 20% psychology students while study 2-4 are mainly psychology students? 

Author Response

Thanks for inviting me to review the article, titled "The effect of situational experiment condition on hasty decision-making in the ‘Beads task’", here are some of the major comments, 

-       For the last sentence of the abstract, “We recommend that testing conditions and incentives are reported.”, is it “should be reported” instead of “are reported” ?

Thanks, you, we have corrected that

-       The authors should state the hypotheses based on the evidence from the existing research in the introduction section

Thank you, we have outlined our hypotheses more clearly, see line 140-142 and 148-149

-       The authors should report the mean age of the participants instead of the range. I just wonder how many of the participants were mature students. It may affect the results found as maturity it’s a potential factor affecting “jumping to conclusion” as well. 

We now report the mean, median, SD and IQR for age. However, we think it unlikely that maturity plays a significant role given the one year difference but large SD per sample. These are not adolescent where we agree that one year may make a difference. If there is an effect of age, it would be through young adults (very young compared to just young) having on average more psychotic-like experiences. However, it is not clear whether these experiences lead to hastier decision making. Please see: https://www.sciencedirect.com/science/article/pii/S0005791618300053 where the authors found not a negative but positive relationship between psychotic-like experiences and draws to decision, i.e., the less delusional-prone the more hasty decision-making

-       Will it be valid to compare the results from the four-study given study 1 only has around 20% psychology students while study 2-4 are mainly psychology students? 

Thank you for this question. Yes, the results are valid. Undergraduate psychology courses are popular and taken by students from various disciplines (engineering, medicine, economy). We do not know how many of the students who took part wanted to become clinical psychologists and how many wanted to have a minor degree in psychology. We refer to all of them as psychology students as we have no information regarding this detail. In addition, previous studies have used (as control group to their patient group) either psychology students or students (from various disciplines). We are not aware that in this information sampling task the topic you study has any effect on task performance.

Reviewer 3 Report

 I agree the article has been improved, but my opinion about the unique contribution this paper provides the field stands.

Author Response

We provide now an even clearer rational for our study in the aims section. We hope this addresses the reviewers concern about the unique contribution to the field.